# Forecasting influenza activity using machine-learned mobility map

Srinivasan Venkatramanan[1], Adam Sadilek [2✉], Arindam Fadikar[3], Christopher L. Barrett [1,4], Matthew Biggerstaff[5], Jiangzhuo Chen[1], Xerxes Dotiwalla[2], Paul Eastham[2], Bryant Gipson[2], Dave Higdon[6], Onur Kucuktunc[2], Allison Lieber[2], Bryan L. Lewis [1], Zane Reynolds[7], Anil K. Vullikanti[1,4], Lijing Wang[1,4] & Madhav Marathe[1,4]

Human mobility is a primary driver of infectious disease spread. However, existing data is limited in availability, coverage, granularity, and timeliness. Data-driven forecasts of disease dynamics are crucial for decision-making by health officials and private citizens alike. In this work, we focus on a machine-learned anonymized mobility map (hereon referred to as AMM) aggregated over hundreds of millions of smartphones and evaluate its utility in forecasting epidemics. We factor AMM into a metapopulation model to retrospectively forecast influenza in the USA and Australia. We show that the AMM model performs on-par with those based on commuter surveys, which are sparsely available and expensive. We also compare it with gravity and radiation based models of mobility, and find that the radiation model's performance is quite similar to AMM and commuter flows. Additionally, we demonstrate our model's ability to predict disease spread even across state boundaries. Our work contributes towards developing timely infectious disease forecasting at a global scale using human mobility datasets expanding their applications in the area of infectious disease epidemiology.

[1] Biocomplexity Institute and Initiative, University of Virginia, Charlottesville, VA, USA. [2] Google Inc., Mountain View, CA, USA. [3] Argonne National Laboratory, Lemont, IL, USA. [4] Department of Computer Science, University of Virginia, Charlottesville, VA, USA. [5] Influenza Division, Centers for Disease Control and Prevention, Atlanta, GA, USA. [6] Department of Statistics, Virginia Tech, Blacksburg, VA, USA. [7] Torc Robotics, Blacksburg, VA, USA. ✉email: adsa@google.com

Seasonal influenza causes significant health impacts (infections, hospitalizations, and deaths) and economic burden (medical care costs and productivity loss) each year worldwide. A recently updated estimate of global influenza burden shows even higher than previously thought magnitude: 291,000–645,000 seasonal influenza-related deaths each year[1]. The 2017/2018 influenza season was of particularly high severity[2]. It resulted in 960,000 hospitalizations in the United States alone and extended periods of high influenza activity. Even with sustained efforts to improve vaccine coverage, influenza continues to strain the healthcare system. This highlights the need for developing systems that provide reliable and relevant forecasts of short-term and seasonal influenza activity. Forecasting seasonal influenza, especially within the United States, has been an area of active investigation in the epidemiological community. Contests such as the CDC Forecasting Challenge[3–6] have fostered innovation and constant information exchange among researchers in the field[7], and[8] provide extensive reviews on the different approaches and methodologies used in practice for forecasting seasonal influenza. In order to forecast the influenza activity characterized by percentage of visits with influenza-like illness symptoms, researchers have used wide-ranging data sources including search trends[9], social media[10], medical claims[11], and weather data[12].

While some of the techniques used to forecast influenza are statistical, i.e., using the patterns in the time-series data of cases and associated datasets, other methods involve a mechanistic representation of the disease process itself. This usually involves capturing the mechanisms and associated factors by which individuals are exposed to the pathogen, infect each other, acquire immunity by recovery or vaccination, etc. In addition to being more descriptive, mechanistic models allow incorporation of interventions and study of potential counterfactual scenarios. Metapopulation models are a popular class of mechanistic models, and are well suited to capture spatial heterogeneity in disease dynamics. Variants of this approach have been successfully used to model and forecast infectious diseases[13–17]. When modeling infectious diseases mechanistically, it is useful to note that in the human population the spread is facilitated by social contacts, which are in turn influenced by the movement of individuals. Researchers have leveraged this fact and have used information on human mobility to predict the dynamics of disease spread. When constructing a model of disease dynamics, even in the absence of high-quality mobility data, one may resort to standard models such as gravity or radiation[18]; provides an extensive review of such mobility models and their wide-ranging applications, including in the field of epidemiology. Datasets that capture movement of individuals at micro and macro scales are increasingly available (Supplementary Table 1), some of them in the public domain enabling their use in infectious disease modeling[19–28] (see Supplementary Notes 1 for a more detailed literature survey). A recent study that combined these different facets and modeled sub-city influenza dynamics for New York City (NYC) was reported in ref. [17]. Similar to our work, they used a metapopulation modeling approach to forecast influenza activity at the borough level (and zip code level) within NYC. Their main focus was to evaluate the presence and absence of travel networks on the forecast performance. In contrast, in our work we tested the effectiveness of different networks derived from official surveys, aggregated location data, and mobility models.

Existing high resolution mobility data are based on call data records and therefore available only in limited jurisdictions, where the telecom provider is operating. As a result, cross-border movement is typically not captured, nor is long-distance international travel. By contrast, the aggregate flows of populations around the world we compute here are based on ambient location that is passively logged by the phones' location sensors, when users opt into this feature. With the aim to better understand global patterns of population movements, we aggregate data from Location History collected passively from smartphones that opted-in to secure location history[29]. Anonymity is an important consideration in our work to ensure that no individual user's journey can be identified, we only share representative models of aggregate data employing differential privacy[30], which intentionally adds noise to the data in a way that maintains both users' privacy, anonymity, and the data's accuracy (see Fig. 1).

In this paper, we use a metapopulation framework to forecast influenza activity in and around the counties (boroughs) of NYC, and further extend the approach for state-level influenza forecasting in Australia. We choose these regions due to the availability of ground truth case data at high spatial and/or temporal granularity. As a source of ground truth, we use data on emergency department (ED) visits for NYC, lab-tested flu positive counts for New Jersey and Australia. To guarantee anonymity and match the resolution of health datasets, AMM data are aggregated at the resolution of counties in the United States, and at the resolution of states for Australia. These anonymized aggregated data come from users who opted-in to share their location data, which is already a vital source of information for

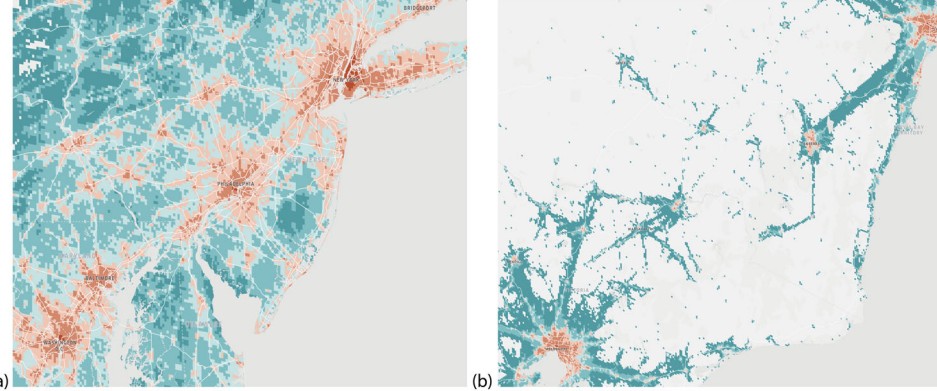

**Fig. 1 Snapshot of AMM. a** Northeastern United States and **b** Southeastern Australia. Color of each 5 km$^2$ cell corresponds to the annual average of total outflow volume in logarithmic scale (warmer colors imply higher connectivity). Note the variations in sparsity and total connectivity. Data are shown in the spatial resolution of cells, although for our subsequent analyses we use AMM aggregated to counties/states to match the surveillance data. Further, for the United States the study is performed at city (New York) and state (NY and NJ) scales, whereas for Australia the study is performed at a national scale, highlighting the versatility of the dataset.

**Table 1 List of datasets.**

| Dataset | Source | Year, Temporal resolution | Spatial resolution |
|---|---|---|---|
| ILI Emergency Department visits in New York City (NYC) | EpiQuery: NYC Syndromic Surveillance | 2016–2017 flu season (daily) | County level |
| ILI Flu-A positive % | CDC FluView | 2016–2017 flu season (weekly) | HHS Region 2, State of New York |
| ILI Lab tested flu positive counts for State of New Jersey (NJ) | The New Jersey Department of Health | 2016–2017 flu season (weekly) | County level |
| Influenza positive counts for Australia (AUS) | National Notifiable Disease Surveillance System, Australia Government, Department of Health | 2016 flu season (daily) | State level |
| Aggregate mobility flows (AMM) | Google | 2016–2017 (weekly) | County level (NY, NJ), State level (AUS) |
| NY, NJ Commuter counts (COMMUTE) | American Community Survey | 2009–2013 (typical day) | County level |
| Interstate commuter flows in Australia | Australian Labor Market Statistics | 2006 census | State level |
| NY, NJ population | U.S. Census Bureau | 2013 population estimates | County level |
| Australia population | Australian Bureau of Statistics | 2016 population estimates | State level |

Each dataset is provided along with the source, temporal, and spatial resolution. The first four datasets pertain to influenza incidence rate monitoring, while the remaining are used to model movement between counties/states. ILI stands for Influenza-Like Illness, which includes influenza and other illnesses that present similar symptoms. Clinical lab tests are used to confirm whether it is influenza, and if so, to identify the particular strain. During a typical influenza season, multiple strains circulate in the population, and Flu-A positive% is the percentage of lab-tested influenza specimens that tested positive for Influenza A. A full list of references are provided in the Data Availability Statement.

estimates of live traffic and parking availability[31]. In addition to AMM, we use commuter counts from the respective Census agencies, unconstrained gravity model, radiation model with a global parameter for commuter proportions, and a no-mobility baseline for comparison purposes. We did not calibrate the gravity model with commuter data (to estimate the exponent, for example), to ensure that the compared networks are independent of each other. The complete list of datasets used in this paper is provided in Table 1, while Fig. 2 provides an overview schematic of the disease simulation and calibration process. More details on the data preparation, disease simulation, and calibration methodology are described in the "Methods".

## Results

**Exploratory analysis of mobility flows.** In an epidemiological model of influenza, the disease transmits via people-to-people contacts, thus a more realistic mobility network might be able to better characterize the epidemic dynamics, leading to better epidemic forecasting of influenza. Before reporting on influenza forecasting performance, we compare the four mobility networks structurally by using the normalized mobility flows for the counties of New York and New Jersey. While comparing the networks in terms of flow distributions and network structure, we focused only on pairwise flows, excluding self-loops, since all four networks had relatively large self-flows (self-loops are omitted only for mobility analysis). We refer to these networks obtained from the anonymized mobility map as AMM, commuter counts from the American Community Survey (ACS) as COMMUTE, gravity and radiation models computed using population sizes and distances as GRAVITY and RADIATION, respectively.

By comparing flow distributions and node betweenness (shown in Fig. 3), we observe that (1) AMM and COMMUTE network are highly positively correlated (Pearson coefficient 0.9) in terms of the flow distribution, while RADIATION model has a reasonable match with AMM (PCC 0.71) and COMMUTE (PCC 0.63); GRAVITY model on the other hand has low similarity (PCC ~0.5) with the other three networks, (2) the mobility networks of AMM and COMMUTE network are sparse and exhibit a community structure usually seen in urban mobility networks, while RADIATION and GRAVITY are dense by

definition, RADIATION network shows similar neighborhood structure to AMM and COMMUTE, (3) AMM and RADIATION networks have similar top betweenness counties, whereas COMMUTE data pick different neighboring counties. Note that while AMM is aggregated and anonymized data sourced from high resolution mobility, COMMUTE is obtained from representative surveys, which may be dated. Similar comparisons for the Australian networks, along with additional details on the mobility data description and network construction are provided in Supplementary Methods and Supplementary Notes 2.

**Forecast performance in NYC.** We first report the networks' ability in predicting ED visits during the 2016–2017 influenza season in the five boroughs of NYC (Bronx, Brooklyn, Manhattan, Queens, and Staten Island). To give some context, nationally the 2016–2017 season had an onset (crossing and staying above baseline for three consecutive weeks) on week 50 of 2016, peaked on week 6 of 2017, and lasted until (stayed above baseline) week 14 of 2017. HHS Region 2 (which includes New York and New Jersey), while having the same peak and end week, technically had an earlier onset (week 47 of 2016). We perform a comparative study in forecasting influenza activity among the mobility networks and the no-mobility baseline by incorporating them in the metapopulation model framework, which we refer to as PatchSim[32]. Each of the instances of the PatchSim with different mobility networks (or no network) is first calibrated using the ground truth observations until a given week (also referred to as data horizon) and the calibrated models are run forward to produce short-term (1–4 weeks look ahead) and seasonal target (onset time, peak time, and peak visit count) predictions for each of the boroughs[33].

This experiment is run retrospectively at different time points of the influenza season serving as the data horizons and comparison is made based on the overall performance of the model throughout the season. We use a Bayesian approach to calibrate the model's disease parameters. The calibration procedure yields a posterior probability distribution on the unknown model parameters, and 90% prediction envelopes are obtained where appropriate. For comparison among the performance of four models, mean absolute percentage error (MAPE) is

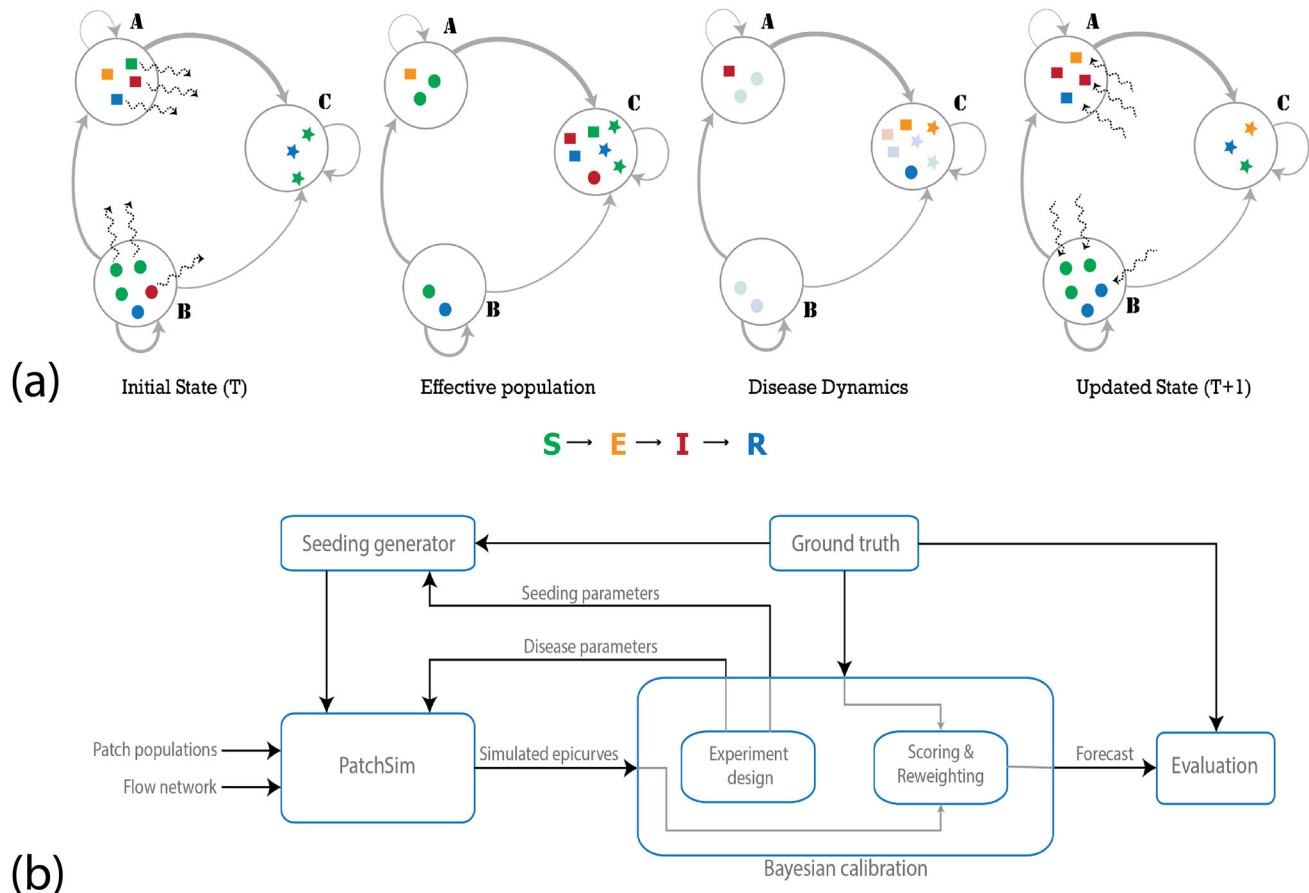

**Fig. 2 Overall methodology. a** Stages in a single iteration of the metapopulation model: the large circles represent the patches in the simulation, and the gray edges represent the travel network, with varying thickness denoting heterogeneity in flow volumes. The sample population shows four individuals in patch A (squares), five in patch B (circles), and three in patch C (stars). The colors represent the disease state the individuals are in (susceptible, exposed, infected, or recovered). The wavy dashed arrows show movement of individuals (randomly chosen according to outgoing edge probabilities). In the first step, individuals are moved from their home patch to another patch (first panel), creating the effective population (second panel). The disease dynamics may include exposure events (transition from S–E), onset of infectiousness (E–I), and recovery events (I–R). The nodes undergoing these transitions are highlighted in the third panel, where we see an onset of infectiousness in patch A, two exposures and a recovery in patch C. Finally, the individuals return to their home patch (fourth panel). Note that, although for descriptive purposes we use individual agents above, the system is actually simulated deterministically using the mass-action principle. **b** Bayesian calibration framework: the calibration involves generating samples from the parameter space, evaluating the corresponding simulated epidemic curves in comparison to the ground truth. The appropriate reweighted epidemic curves are combined to provide the forecast. See Supplementary Methods for details on both the disease model and the calibration procedure.

computed for each data horizon based on 1-to-4-week-ahead median predictions.

Figure 4a, b shows the 90% envelopes calibrated and forecasted ED visits predictions by each of the models in five NYC counties at data horizons of week 2 (pre-peak) and week 6 (post-peak) of 2017, respectively. From Fig. 4c, it can be seen that the performances of AMM and commute are similar, and are better than gravity and baseline models in terms of overall MAPE. We choose to show the short-term prediction performance between week 50 of 2016 and week 12 of 2017, since they roughly coincide with the onset and end of the 2016–2017 influenza season. The no-mobility baseline performs better during early weeks, especially before onset, but as the season progresses, its performance deteriorates in comparison to the other network models. The MAPE performance for individual boroughs and two other seasons (2015–2016 and 2017–2018) are provided in Supplementary Notes 2. We note that while the ordering of performance varies between seasons, AMM, COMMUTE, and RADIATION consistently perform similar to each other.

Figure 5 shows the predictive distribution of the seasonal forecasts (specifically peak time) provided by different models at different data horizons for each of the boroughs (other seasonal targets such as peak intensity and onset time are provided in Supplementary Notes 2). We note again that as the data horizon increases, the models produce tighter estimates of the seasonal forecasts. At the beginning of the season, all mobility models predict an early peak, but as the season evolves the prediction intervals get narrower and cover the ground truth. We also note that for the 2015–2016 and 2017–2018 season (see Supplementary for figures), similar conclusions can be drawn, where AMM, COMMUTE, and RADIATION perform similarly in terms of predicting the seasonal targets.

**Extending beyond the NYC boroughs**. In addition to estimating influenza activity into the future, the mechanistic model can also be used to meaningfully impute potential gaps in case data in neighboring regions, utilizing the available case data and inter-regional mobility. To test the utility of the disease model along with the higher resolution AMM for this purpose, we first perform a leave-one-out cross validation study, where each time the PatchSim is calibrated using partial ground truth data

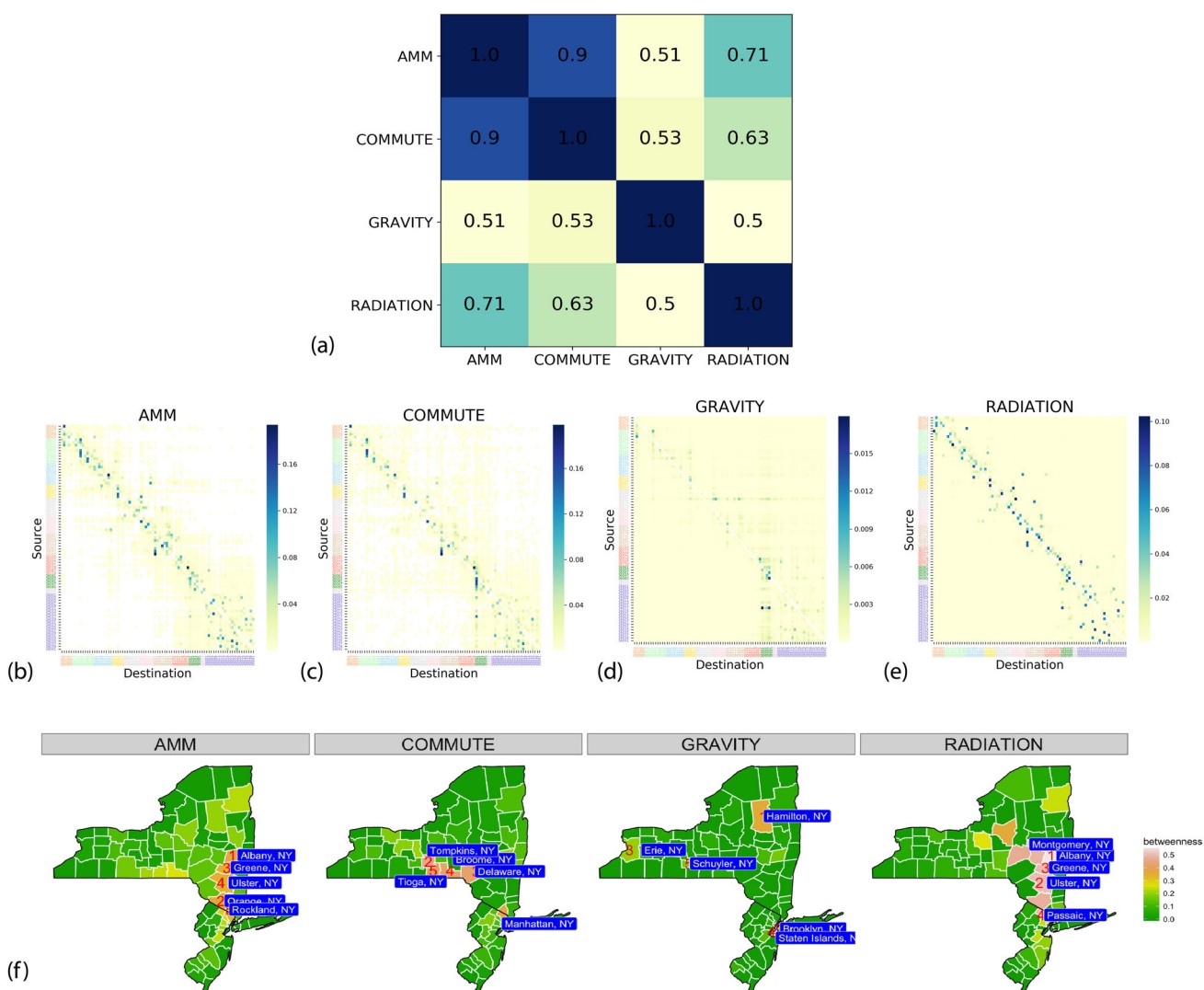

**Fig. 3 Structural comparison of the networks. a** Correlation between networks. Pearson correlation coefficient between the different mobility networks (FIPS-sorted by source and destination). AMM and COMMUTE have a high positive correlation value (0.90), followed by RADIATION model (0.71 with AMM), while GRAVITY has relatively small positive correlation with the other networks. **b–e** Adjacency flow matrices of AMM, COMMUTE, GRAVITY, and RADIATION. Nodes are arranged by spatial proximity. Heatmap color is indicative of normalized flow volume (darker color denotes larger flow). Note that while AMM and COMMUTE are sparser and seem to be clustered, GRAVITY network, by virtue of its definition, is more homogeneous. More detailed versions of AMM and COMMUTE flows are shown in Supplementary Figs. 3 and 4. **f** Betweenness measures at county level for all four networks. Among the four networks, AMM, and RADIATION models have similar sets of top counties by betweenness, while COMMUTE chooses nearby but distinct counties. These counties are reasonable choices, since they connect rural counties, e.g., Allegany or St. Lawrence or Cape May, to highly urbanized counties like Manhattan or Brooklyn. However, in GRAVITY we failed to identify any such distinct pattern.

leaving one of the five counties unused, and the calibrated model is used to predict the epicurve of the left-out county. Figure 6 shows the 90% prediction intervals for the held out counties along with peak time and peak intensity predictions. The results suggest that such a setup can be used to forecast even in regions where no past case data are available, provided we have case data for well-connected regions and a good estimate of inter-regional mobility.

As further proof of these idea, we extend the radius of the PatchSim simulation model to cover the entire states of New York and New Jersey. Based on past ILI data, while New York and New Jersey have typically peaked within 2 weeks in the last three seasons, in general, their peak timings could be off by about 8 weeks. Further, the county-level influenza dynamics also shows sufficient variability, and hence it would be interesting if our model could capture it. The disease model is again calibrated only

for the NYC counties using the ED visits data, and the calibrated model is used to recreate the influenza season (ED visits to be exact) for each county in NY and NJ state. We show in Fig. 7 the simulated trajectories and the corresponding ground truth for the five boroughs and the six neighboring counties in New Jersey. Note that, since we did not have ED visits data for New Jersey, we used the lab-tested positive counts made available by the state's health department. We also report summary statistics such as onset time, peak time, and end of the season, to show that the simulated trajectories are quite close to recreating the flu season in these neighboring counties. We also tried to match the aggregate curves to ILI% incidence from NY state and HHS Region 2 (comprising New York and New Jersey) with limited success. The simulated curves for all NY and NJ counties with ground truth data, and the state/region level fits are shown in Supplementary Notes 2.

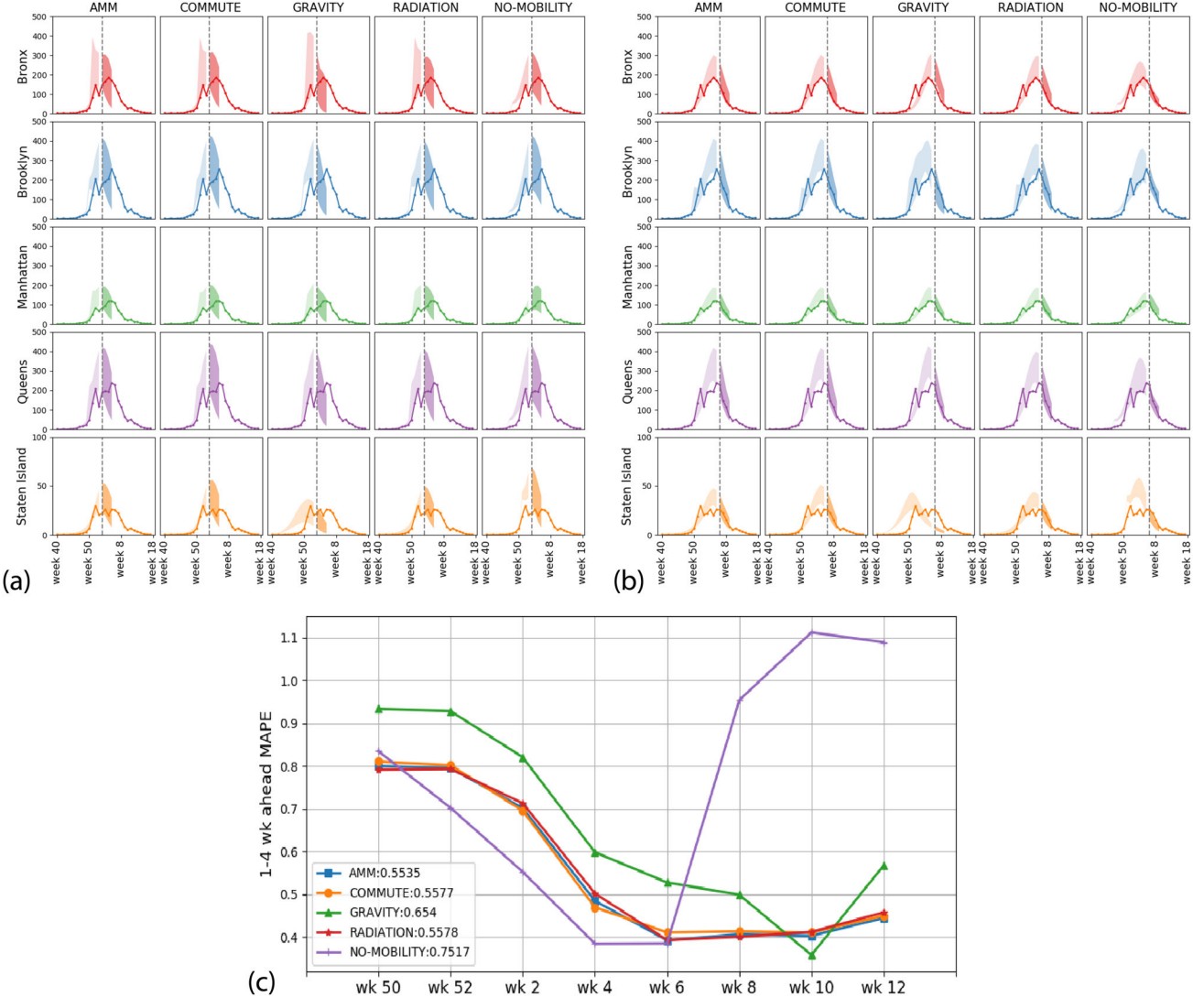

**Fig. 4 Prediction envelopes and comparative forecast performance.** The figure shows the comparative performance of the five models in predicting emergency department visits in the five boroughs of New York City. **a**, **b** show the calibration envelope (light shaded) and 90% prediction envelope (dark shaded) at weeks 2 and 6 of 2017, respectively. These weeks were chosen to be roughly two weeks pre-peak and two weeks post-peak across boroughs. The solid lines show the ground truth, and colors throughout are representative of boroughs. The plots are arranged horizontally by boroughs and vertically by the mobility network. **c** MAPE performance. One-to-four-week-ahead median predictions at different data horizons are used to evaluate the mean absolute percentage error (MAPE), for every other week of the flu season. Note that the lower the MAPE, the better the network. The overall MAPE is provided in the legend shows that AMM, COMMUTE, and RADIATION models perform similarly, and do better than the GRAVITY or NO-MOBILITY baselines.

**Forecasting influenza in Australia**. To demonstrate the generality and potential global scope of AMM, we chose to replicate our approach for seasonal influenza forecasting in Australia. We chose Australia for the following reasons: (1) sparser populations spread across a wider spatial scale, where the effects of mobility may be more pronounced[34], (2) the presence of high-quality surveillance data in the public domain, (3) "inverse" influenza season in the Southern Hemisphere. As a test case, we evaluated short-term forecasts for the 2016 influenza season. Similar to the NYC study, we compared the performance of AMM to that of COMMUTE, GRAVITY, RADIATION, and a NO-MOBILITY baseline. Data sources for commuter flows, populations, and influenza surveillance are listed in Table 1 (data preparation details are in the Supplementary Methods). Figure 8 summarizes the underlying datasets and the forecast performance comparison among the mobility models. As earlier, we note that AMM performs on-par with RADIATION, COMMUTE data. Compared to the NYC results, we find little difference between NO-MOBILITY and other mobility models, perhaps due to the sparsity of the region. The calibrated and forecast curves along with MAPE per region are provided in Supplementary Notes 2.

## Discussion

Infectious disease forecasting has risen to prominence in the recent decade, thanks to the increased availability of public health surveillance system data and development of sophisticated methodologies. Real-time disease forecasting is still plagued by the lack of current estimates on mobility and interaction patterns, which are known to be key drivers of disease spread. We demonstrate the utility of high-quality mobility data for disease modeling and forecasting in areas that have detailed and extensive ground truth available, in order to be able to exactly quantify the efficacy of our approach.

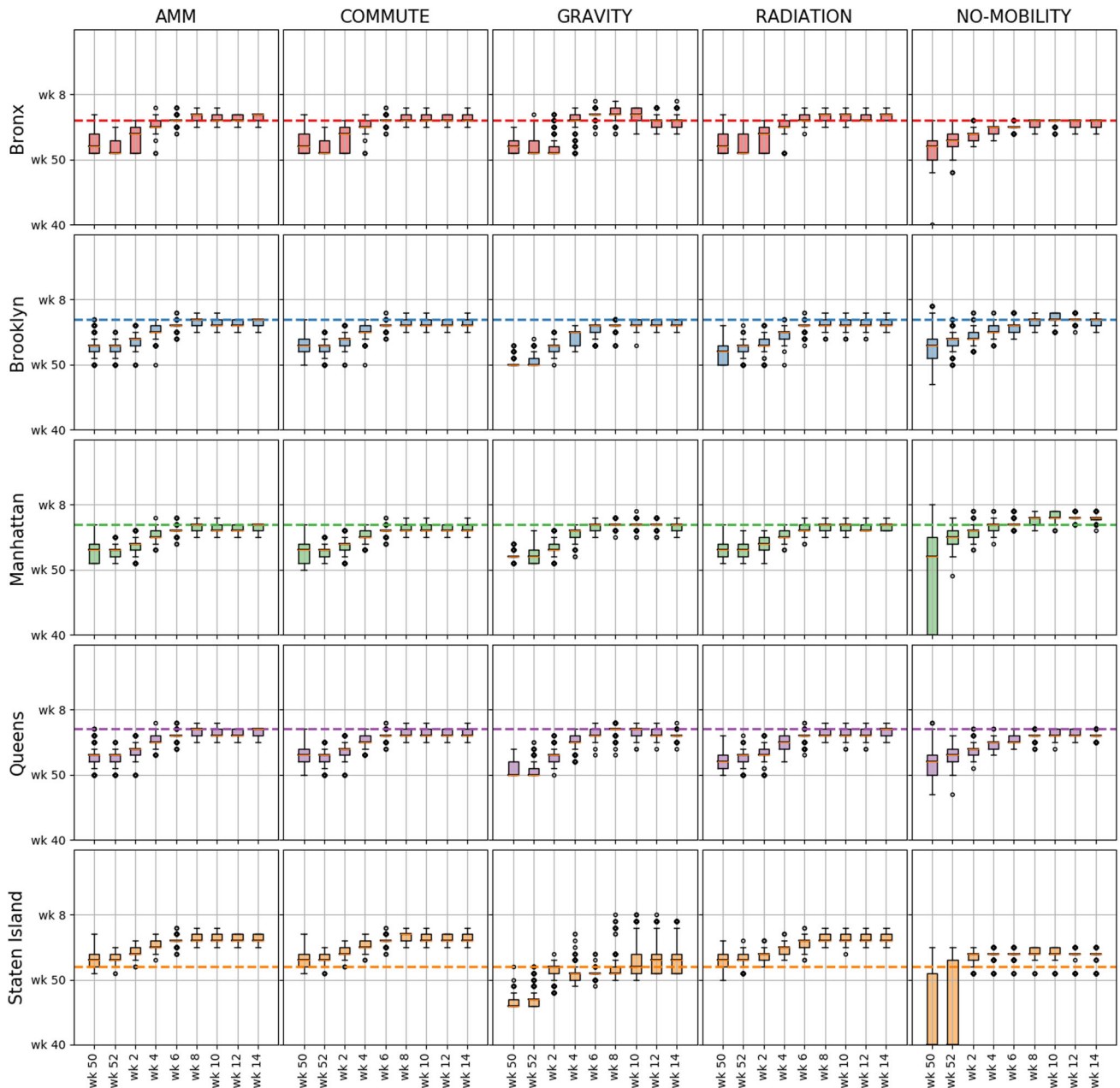

**Fig. 5 Multi-horizon comparison of seasonal forecasts across networks.** For each of the networks (columns), and each borough (rows), the boxplots (median, IQR, and whiskers at 1.5 IQR) show the posterior predictive distribution for peak time forecast at different time horizons ($n = 1000$). Similar plots for other targets (peak value, onset time) and other seasons are available in Supplementary Notes 2.

However, the implications of our approach extend beyond high-resource parts of the world. In fact, most regions, even in high-resource countries, do not have disease case data at sub-state resolutions. Another drawback of traditional case data streams is the temporal lag in data availability, thus making even nowcasting the current state of influenza a practical challenge. Models based on population accounting, both static (census counts) and dynamic (commuter and migration flows) are neither comprehensive nor current in most parts of the world, thus hindering the performance of mobility models used as proxies. In contrast, mobility statistics such as AMM are more timely and high-recall, and therefore serve as an important complement augmenting and filling gaps in influenza data. Given the coverage, the forecasting framework can be expanded to make predictions at a global scale. To this end, we apply our model to Australian flu epidemics, in

addition to the United States. The results show consistent performance in a different region of the world with a more sparse population, different flu dynamics, weather, and seasons. While quite useful in the context of seasonal influenza, such a global system becomes especially relevant and essential in the context of emerging infectious diseases where mobility plays a stronger role in quantifying case introduction risk.

Measures of flows of population at a fine granularity both in space and time have been limited due to lack of timeliness, availability, and accuracy of observational data. Here, we show that anonymous and aggregated mobility data improve the quality of models and can unlock interesting approaches. Crucially, we demonstrate this method is applicable at scale and is not limited to select jurisdictions and geographical areas that happen to collect necessary survey-based data at considerable expense

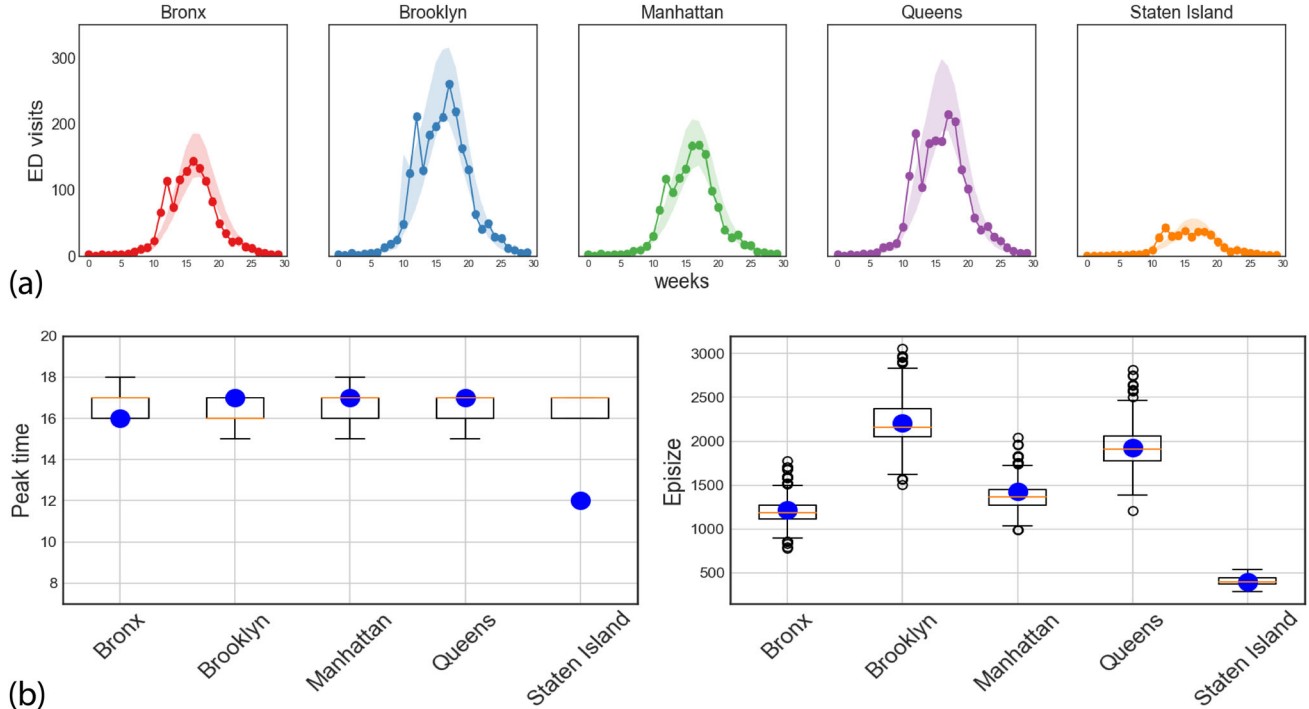

**Fig. 6 Leave-one-out cross validation performance. a** For each of the boroughs, the ground truth is shown in solid line along with the 90% calibrated envelope when the borough's data were left out of the calibration process. **b** Boxplots (median, IQR, and whiskers at 1.5 IQR) of the predicted peak time and total ED visits (episize) for the left-out borough ($n = 1000$), with the ground truth shown by blue dots.

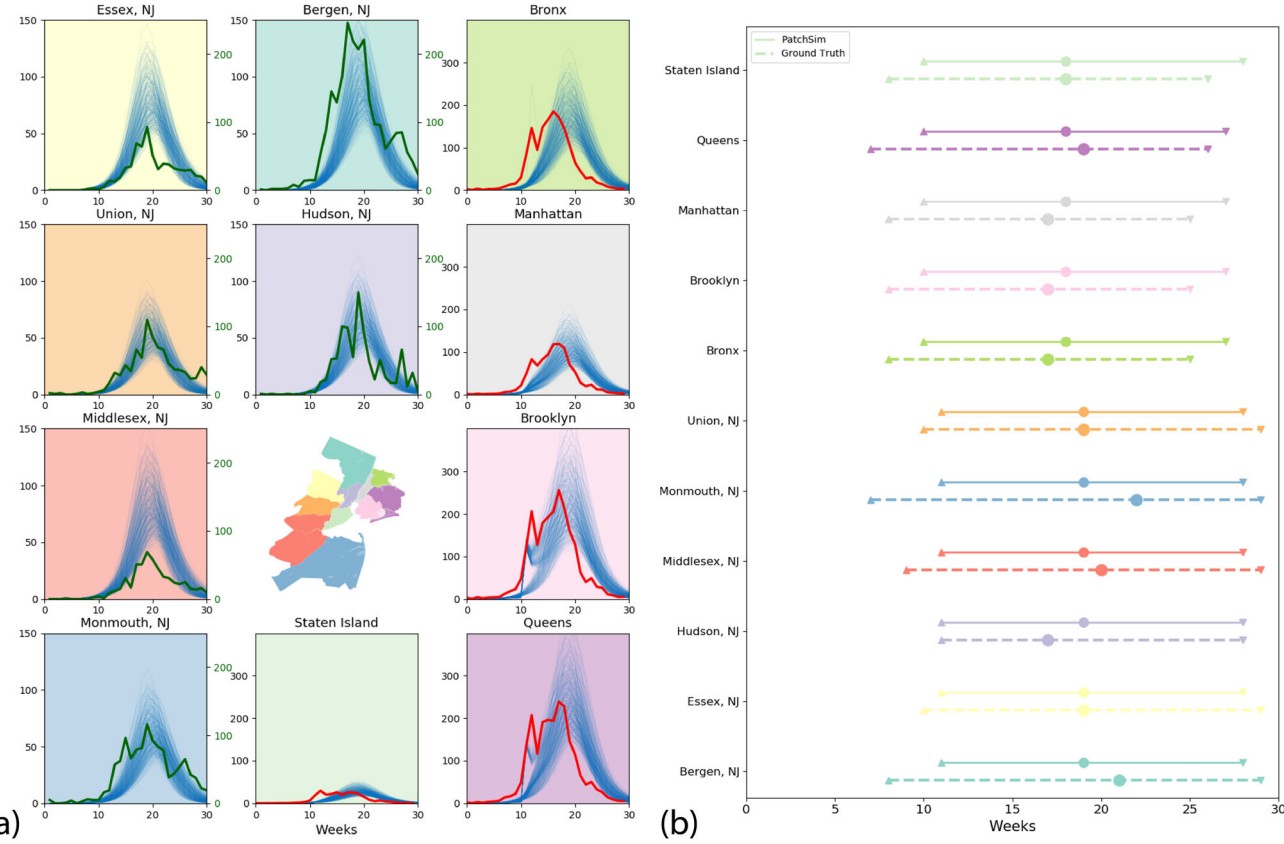

**Fig. 7 Extrapolation for New Jersey counties adjacent to NYC. a** Simulated trajectories (thin lines) along with ground truth (bold lines) shown for the five boroughs of New York City, and the six neighboring counties in New Jersey. The plots are color-coded to match the respective county in the inset map. **b** Comparison of summary statistics (onset time, peak time, and end of season) for each of these counties. Onset and end of season is defined with respect to a threshold of 10% of peak value.

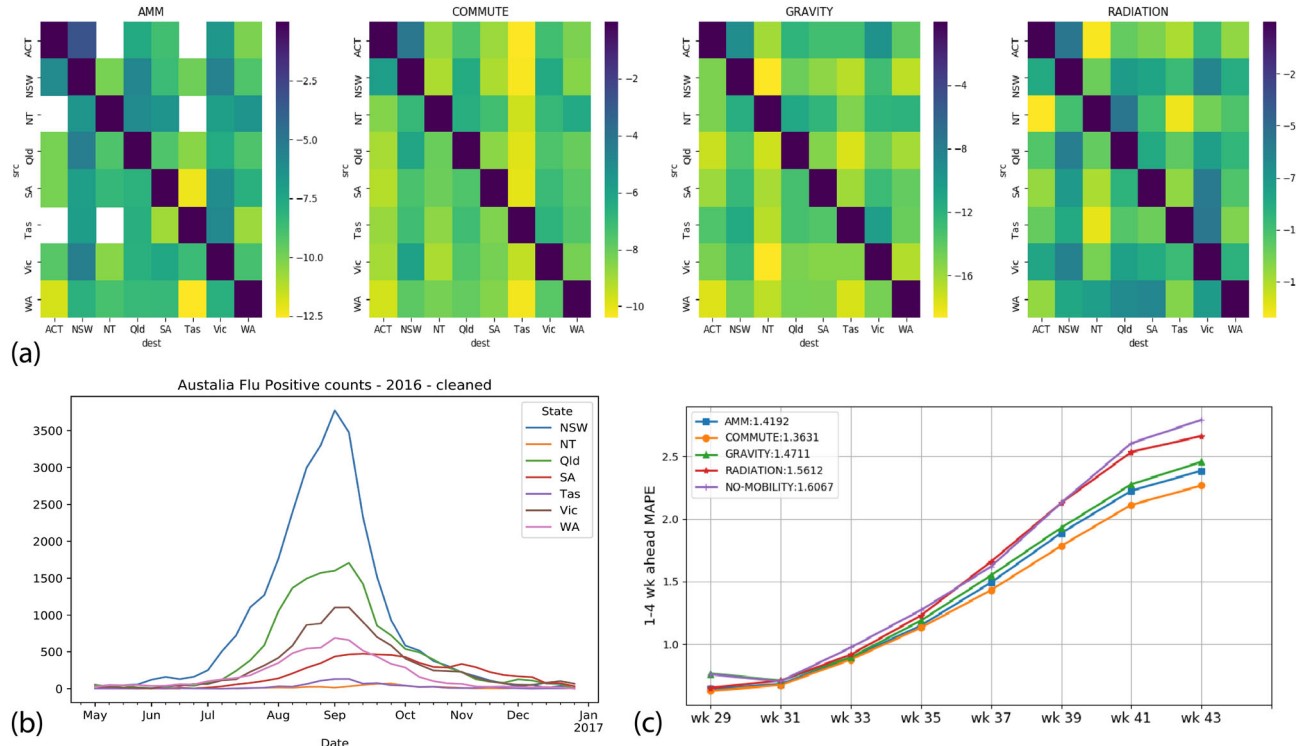

**Fig. 8 Summary of results for Australia. a** Mobility datasets for Australia: the heat maps show a comparison between the different mobility networks used for Australia at the state level. The reported values are the natural logarithm of the normalized flows. Note that while there is a significant self-loop component in all four networks, there is a higher similarity in magnitude and distribution between COMMUTE and AMM. **b** Influenza surveillance for Australia, 2016: the dataset obtained from Australia's National Notifiable Disease Surveillance System (NNDSS) shows flu positive counts at State level aggregated to weekly resolution, with the baseline count removed (see Supplementary Methods for details). **c** Forecast performance comparison: similar to Fig. 4c, this figure shows the average MAPE across weeks for the four mobility networks (including the no mobility baseline). We see that all networks have similar performance with COMMUTE and AMM being the top two models. We also note that the NO MOBILITY baseline is not very different in performance, highlighting the sparsity of the region.

and time delays. Our model captures global flows of populations and is not limited to within-state movements or other geographical boundaries.

Methodologies based on aggregated samples of mobility do have their limitations. For instance, based on the way the trips are defined, it is difficult to distinguish between trips with varying dwell times. In the context of infectious diseases, the duration an individual spends at a location has a significant effect on the likelihood of passing on or contracting the infection. Further, due to anonymization, the datasets do not distinguish between individuals who reside in the patches, and those who are transients (e.g., tourists). The mobility data need to be appropriately aggregated to account for mobility patterns that have a significant impact on disease spread. Identifying the right aggregation mechanism for a given disease model is still an open challenge. We also want to note that, comparing four different mobility networks via epidemic modeling and prediction is not immune to effect confounding, since it is hard, if not impossible to distinguish the effect of different networks on the predictions from the effect of our choice of disease model and the calibration technique. Further, it is possible that a combination of aggregated mobility traces with sophisticated data-driven models might yield superior performance compared to either one in isolation. Recent collaborative efforts[3] have revealed that ensemble approaches tend to perform better in forecasting, thus motivating further research on effectively combining these models. Finally, the availability of good quality ground truth is necessary to test different approaches in incorporating such datasets.

The system being considered in the paper is not closed in reality, due to flows from/to neighboring states or due to international air travel. To look at the impact on neighboring states, we have modeled the NY–NJ states (HHS Region 2). Modeling disease dynamics for any region poses this "open world" problem. Finding correction terms, and to do so in a data-driven fashion using incoming flows and influenza indicators in external regions would be a broad open question worth pursuing.

The Google Aggregated Mobility Research Dataset (referred here for brevity as anonymized mobility map, or simply mobility map) contains anonymized mobility flows aggregated over users who have turned on the Location History setting. Therefore, our mobility map shows only a sampled view into the actual population movements. However, our model makes the assumption that this is fairly representative of the aggregate inter-regional mobility patterns[18,35–37]. As recent third party surveys indicate[38], Android has a fairly uniform coverage ~60% across gender, age, and other demographic factors. Further, we consider our work as a complement to alternative techniques, since any mobility model derived from real-world data (for instance commuter flows) will suffer from some form of bias (participation bias in census). Finally, noise is strategically added to the data to preserve privacy, as described in the "Methods". The amount of noise is designed to protect individuals' privacy while not significantly distorting the aggregate statistics. The mobility flows our system computes protect privacy and anonymity by leveraging differential privacy algorithms combined with data aggregation over large geographical areas and time intervals.

Beyond the scope of our current work, we believe the mobility data and the framework can be extended to make predictions at finer spatial (e.g., zip code) and temporal (daily) resolutions. The scalable nature of AMM can help in areas that cannot invest as much in surveys and infrastructure as NYC. The global nature of the data source makes it an ideal candidate for potential pandemic preparedness studies and rapid risk estimation during an unfolding outbreak. With sophisticated agent-based models[39] and on-device learning[40,41], such techniques could also lead the way towards individual-level forecasting without involving the same privacy and anonymity constraints.

## Methods

**Anonymized mobility map (AMM)**. The Google Aggregated Mobility Research Dataset contains anonymized mobility flows aggregated over users who have turned on the Location History setting, which is off by default. This is similar to the data used to show how busy certain types of places are in Google Maps—helping identify when a local business tends to be the most crowded. The dataset aggregates flows of people from region to region.

To produce this dataset, machine learning is applied to logs data to automatically segment it into semantic trips[42]. To provide strong privacy guarantees, all trips were anonymized and aggregated using a differentially private mechanism[43] to aggregate flows over time (see ref. [44]). This research is done on the resulting heavily aggregated and differentially private data. No individual user data were ever manually inspected, only heavily aggregated flows of large populations were handled.

The automated Laplace mechanism adds random noise drawn from a zero mean Laplace distribution and yields $(\epsilon,\delta)$-differential privacy guarantee of $\epsilon = 0.66$ and $\delta = 2.1 \times 10^{-29}$ per metric. Specifically, for each week W and each location pair (A, B), we compute the number of unique users who took a trip from location A to location B during week W. To each of these metrics, we add Laplace noise from a zero mean distribution of scale 1/0.66. We then remove all metrics for which the noisy number of users is lower than 100, following the process described in ref. [43], and publish the rest. This yields that each metric we publish satisfies $(\epsilon,\delta)$-differential privacy with values defined above. The parameter $\epsilon$ controls the noise intensity in terms of its variance, while $\delta$ represents the deviation from pure $\epsilon$-privacy. The closer they are to zero, the stronger the privacy guarantees. Each user contributes at most one increment to each partition. If they go from a region A to another region B multiple times in the same week, they only contribute once to the aggregation count. No individual user data was ever manually inspected, only heavily aggregated flows of large populations were handled.

We aggregate flows within the US spatially at county level and temporally at week level to obtain the mobility map. AMM contains normalized flows between pairs of counties in each week from 2016 week 40 to 2017 week 39, where weeks are indexed from week 00 to week 52 in a calendar year. $\frac{U_{t,ij}}{C}$, where $U_{t,ij}$ is the number of unique users making a trip from county $i$ to county $j$ in week $t$, and $C$ is an undisclosed constant larger than the maximum flow over the entire year $C > \max_{t,i,j} U_{t,ij}$. This dataset covers most counties (3099) in the USA except those in Hawaii and DC. For the purpose of the paper, we used data pertaining to counties in New York and New Jersey, and at state level for Australia. In each study, flows connecting the regions of interest to the outside were not included.

**Mobility data preparation**. We construct mobility networks (i.e., normalized flows between counties/states) based on various mobility datasets, including AMM, the Commute flow data obtained from the ACS, gravity, and radiation models of mobility. For any region, e.g., NYC, we generate a directed weighted network where a node represents a county and a directed edge represents a flow from a source county to a destination county. The edge's weight is defined as the normalized flow (i.e., the outgoing flows of each node sum to 1) coming from the underlying mobility dataset. (1) AMM: the weight is the normalized Google mobility flows averaged across weeks from 2016 week 40 to 2017 week 39. (2) COMMUTE: the weight is the normalized commuter counts from source to destination obtained from ACS 2009–2013. In addition to the reported self-loop, we add the non-commuter population which is calculated by subtracting all commuter counts from population size of the source county. (3) GRAVITY: the weight is the normalization of gravity flows calculated as $\frac{P_i P_j}{(d_{ij}+1)^2}$ where $P_i, P_j$ represent the population sizes (US Census, 2013 population estimates) of county $i$ and $j$, and $d_{ij}$ denotes the distance between $i$ and $j$ computed as the great-circle distance between the county centroids. (4) RADIATION: Using the definition in ref. [35], the flow for $i \neq j$ is obtained as $T_i \frac{P_i P_j}{(P_i + P_j + S_{ij})(P_i + S_{ij})}$ where $S_{ij} = \sum_{k: d_{ik} < d_{ij}} P_k$ is the population living in the circle centered around $i$ with radius $d_{ij}$. $T_i$ is the total commuter outflow from each patch, and is modeled as $T_i = \gamma P_i$, with $(1 - \gamma) P_i$ set as the self-loop flow. For NYC and NJ experiments, based on US commuter data analysis in ref. [35], we set $\gamma = 0.11$. These flows are then normalized to be compatible with the simulation model. The mobility networks are constructed for both NYC (consisting of five counties) and a region of two states, New York plus New Jersey (consisting of 83 counties).

We adopted a similar approach to obtain the AMM, COMMUTE, GRAVITY, and RADIATION flows for Australia. While in NYC we simulated at the level of boroughs (counties), for Australia, we chose to simulate at the spatial scale of states, based on surveillance data availability and also to showcase the generality of the AMM dataset. Interstate commuter flows were obtained from the Australian Labor Market Statistics based on the 2006 Census data. For the RADIATION model, based on median commuter outflow ratio to population sizes, $\gamma$ was set to be 0.004.

To compare the different networks, we used pairwise correlation and betweenness centrality of the nodes in the network. The correlation was computed as the Pearson correlation coefficient between the flattened flow matrices (i.e., vectors). It was used to show the similarity (or lack thereof) between two flow matrices. We used the definition of betweenness for a weighted network (fraction of pairwise weighted shortest paths passing through a node)[45]. The inverse of the normalized flow between a pair of nodes is used as the edge weight. Betweenness centrality is known to be one of the most effective heuristics in controlling epidemics on networks[46]. Although the relationship to a metapopulation model is not evident, betweenness is a useful measure to capture critical counties for the mobility flow. Additionally we calibrated the gravity model separately to the AMM and COMMUTE datasets of New York plus New Jersey, and tested the temporal matrices obtained from AMM for stationarity (see Supplementary Note 2).

**Case data preparation**. The case data used in this work include: (1) NYC ILI ED visits provided by the NYC Department of Health. It contains daily ED visits for ILI per county within NYC for the past three seasons. The daily ED visits are aggregated to weekly data and scaled by the influenza virus isolation rates (aka percent positive, provided by WHO-NREVSS clinical labs) to obtain the ILI+ epidemic curves. We use the isolation rates corresponding to HHS Region 2, which includes NYC. (2) NJ Flu positive counts provided by the NJ Department of Health. It is a weekly cumulative total positive specimens per county for the past three seasons (week 40 of a particular year to the following year's week 20). We calculate the weekly newly identified isolates by subtracting the cumulative count of previous week from that of the current week. (3) ILI% for NY state and HHS2 region provided by the Centers for Disease Control and Prevention (CDC). It is the total number of visits for ILI over total patient visits for the past three influenza seasons. (4) Laboratory confirmed influenza for Australia Influenza surveillance data was obtained from the National Notifiable Disease Surveillance System (NNDSS) maintained by the Australian Government Department of Health, and aggregated to weekly resolution for May–December 2016. The public dataset contains notification data collected on laboratory confirmed influenza via NNDSS at weekly resolution, for the states (excluding Australian Capital Territory), classified by type/subtype, age, sex, etc. We computed the total influenza positive counts per week and removed the baseline (the minimum count for each state in the year) to obtain the ground truth for the metapopulation model.

**Metapopulation model**. PatchSim is a metapopulation SEIR model simulated using difference equations. From metapopulation modeling terminology, patches are habitable units (e.g., spatial regions) within which homogeneous mixing of individuals is assumed. For instance, in the NYC study, the individual boroughs (five of them) are the patches, whereas in the Australia study, the eight states (including Northern Territory and Australian Capital Territory) are modeled as separate patches.

Given a set of patches $\mathcal{N}$ to denote spatial regions (for example, counties in NYC or states in Australia), associated with each patch $i$, we have population $P_i$, and state tuple $Z_i(t)$ denoting the number of individuals in each of the disease states at time $t$. For a typical SEIR (Susceptible → Exposed → Infected → Recovered) model, the set of states is given by $\mathcal{Z} = \{S, E, I, R\}$, with $\sum_{z \in \mathcal{Z}} z_i(t) = P_i$. Between a pair of patches $i$ and $j$, we have the flow $F_{ij}$ denoting the fraction of individuals belonging to home patch $i$ spending their day in away patch $j$. In order to conserve patch populations (i.e., commuting model), we assume $\sum_{j \in \mathcal{N}} F_{ij} = 1$. The mobility is assumed to be homogeneous and memory-less, i.e., the commuting individuals according to $F_{ij}$ are assumed to be picked at random from the population $P_i$ independent of their disease state, and independently for each day of the simulation. Due to the movement of individuals, the effective population of patches may differ from their home population $P_i$. This in turn also affects the state tuple $Z_i$.

PatchSim steps through the disease simulation in daily epochs. In order to compute the change in state tuple $\Delta Z(t) = Z(t + 1) - Z(t)$, it incorporates (1) movement of individuals from their respective home patches to away patches according to $F_{ij}$, (2) exposures, infections, and recoveries happening in the away patches, and (3) integration of state updates at the home patches. Let $\beta$ represent the probability of exposure per day per SI contact, $\alpha$ the infection rate and $\gamma$ recovery rate. $\alpha$ can be thought of as the reciprocal of mean incubation period, and $\gamma$ the reciprocal of mean infectious period. Thus, given the disease parameters $(\beta, \alpha, \gamma)$ and a seeding profile $X$, PatchSim uses the population vector $P$ and flow matrix $F$ to produce the spatio-temporal evolution of disease states $Z$. The exact equations are provided in the Supplementary Methods section with the code, software documentation, and model description available at ref. [32].

**Bayesian calibration**. Model calibration is the process of estimating the unknown parameters of the model with the help of observed data. In the context of our

disease simulation PatchSim, we will be estimating the disease parameters and seeding profile by calibrating it against observed ground truth of influenza incidence. We adopt a Bayesian approach to calibrate the PatchSim model, where we begin with a prior distribution on the unknown parameters, which are then combined with the likelihood of observing the data to produce the posterior distribution on the parameter space.

We begin by defining a statistical model for the observed data as a noisy version of model output, usually Gaussian, independent and identically distributed across the data points. The likelihood of observing the ground truth, given the model is run with parameter $\theta$ can then be written as a multivariate Gaussian across time points and patches. Given the prior distribution $\pi(\theta)$ and the data likelihood $L(y|\theta)$, the posterior distribution can be written by Bayes' theorem.

The analytic solution of the posterior distribution is often not feasible because of the complex simulation model, and hence Monte Carlo approaches to explore the posterior space are often used in such situations. Especially, in our context, we use importance sampling to generate realizations from the posterior distribution. Our choice of importance distribution is the prior $\pi(\theta)$ itself. This reduces the calculation of importance weights $\omega$ to just computing the data likelihood $L$ at each sample from the prior. Thus a re-sample $\hat{\theta}$ from the original set of parameters $\theta$ with probabilities proportional to $\omega$, with replacement, constitute a sample from the posterior distribution. The calibrated forecast can then be produced by running the PatchSim model at the parameter values $\hat{\theta}$, which are then used to compute several summary statistics on the forecast. More details on the calibration framework, and its adaptation for PatchSim and influenza forecasting is described in the Supplementary Methods.

**Reporting summary**. Further information on research design is available in the Nature Research Reporting Summary linked to this article.

## Data availability
Emergency department visits in New York City (NYC) related to Influenza-like Illness were obtained from NYC EpiQuery Syndromic Surveillance portal (https://a816-healthpsi.nyc.gov/epiquery/). Influenza-A positive percentages were obtained from US CDC FluView (https://gis.cdc.gov/grasp/fluview/fluportaldashboard.html). Lab tested flu positive counts for New Jersey were obtained from NJ State Department of Health webpage (https://www.nj.gov/health/cd/statistics/flu-stats). Influenza positive counts for Australia were obtained from Australian Government Department of Health's website for National Notifiable Diseases Surveillance System (http://www9.health.gov.au/cda/source/pub_influ.cfm). Commuter flows for New York and New Jersey were obtained from American Community Survey (https://www.census.gov/data/tables/time-series/demo/commuting/commuting-flows.html). Interstate commuters for Australia were obtained from the Australian Labour Market Statistics (https://www.abs.gov.au/AUSSTATS/abs@.nsf/Previousproducts/6105.0Feature%20Article1Oct%202008). County population sizes for NY and NJ were obtained from US Census Bureau (https://www.census.gov/topics/population.html). State and territory population sizes for Australia were obtained from Australian Bureau of Statistics (https://www.abs.gov.au/statistics/people/population). Preprocessed versions of the above datasets used in the simulation are provided in the code repository (https://github.com/NSSAC/AMMFluForecasting). The Google Aggregated Mobility Research Dataset used for this study is available with permission from Google LLC.

## Code availability
The simulation engine (PatchSim) for the metapopulation model is available at ref. [32], with appropriate documentation and example datasets. The custom codes used for calibration, forecasting, and evaluation are available at ref. [33].

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

## Acknowledgements

We thank Rishi Bal, Avi Bar, Curt Black, Susan Cadrecha, Stephanie Cason, Ciro Cattuto, Charina Chou, Katherine Chou, Iz Conroy, Liz Davidoff, Jeff Dean, Jutta Degener, Damien Desfontaines, Jason Freidenfelds, Vivien Hoang, Sarah Holland, Michael Howell, Pan-Pan Jiang, Ali Lange, Bhaskar Mehta, Caitlin Niedermeyer, Genevieve Park, Chase Rigby, Kathryn Rough, Flavia Sekles, Calvin Seto, Rachel Soh, Aaron Stein, Chandu Thota, Michele Tizzoni, and Ashley Zlatinov for their insights and guidance. We also thank our external collaborators and members of the Network Systems Science and Advanced Computing Division (NSSAC) for their suggestions and comments. This work has been partially supported by DTRA CNIMS Contract HDTRA1-11-D-0016-0001, NIH MIDAS Grant 5U01GM070694, NIH Grant 1R01GM109718, NSF DIBBS Grant ACI-1443054, NSF EAGER Grant CMMI-1745207, NSF BIG DATA Grant IIS-1633028. The findings and conclusions in this report are those of the authors and do not necessarily represent the official position of the funding agencies or Centers for Disease Control and Prevention.

## Author contributions

S.V. contributed to design, problem formulation, analysis, experiments, and writing. A.S. contributed to problem formulation, design, data acquisition, analysis, and writing. A.F. contributed to analysis, experiments, and writing. C.L.B. contributed to design and writing. M.B. contributed to writing. J.C. contributed to design, analysis, and writing. X.D. contributed to data acquisition and writing. P.E. contributed to writing. B.G. contributed to data acquisition. D.H. contributed to analysis, experiments, and writing. O.K. contributed to data acquisition. A.L. contributed to data acquisition. B.L.L. contributed to design, analysis, and writing. Z.R. contributed to design, problem formulation, and analysis. A.K.V. contributed to design and writing. L.W. contributed to analysis, experiments, and writing. M.M. contributed to problem formulation, design, analysis, and writing.

## Competing interests

The authors declare no competing interests.
