## [Peer Review File · Nature Communications]

Reviewers' comments:

Reviewer #1 (Remarks to the Author):

In this study the authors present the results of a modelling approach to forecasting seasonal influenza activity in the USA, based on the integration of mobility data with a metapopulation dynamic model.

Overall, this is an important piece of work that clearly has its merits in the field of digital epidemiology.

The most significant and novel contribution of this work stands in using high-resolution mobility maps obtained through the analysis of large-scale position data available to Google through its Android user base. As shown in Figure 1, this is a powerful data source which can provide information on human movements at the global scale, with a resolution that is not typically available through traditional data sources.

While the opportunity of integrating such data into influenza forecasts is clearly exciting, the actual benefit of using such data is not so clear from the results presented in the study.

In particular, the claim that "aggregated mobility data significantly improves the quality of models" does not seem to be supported by the results, while instead appears that the Global Mobility Map and the commuting data have a nearly identical performance across all forecasting horizons.

One reason for this, in my opinion, is that the study focuses on one of the most densely populated regions of the world. It is natural to expect disease patterns to be highly synchronized among the boroughs of New York City. Indeed, it would be reasonable to consider the whole New York City as a single patch in the metapopulation model.

High-resolution mobility data could be much more valuable, in principle, to forecast the spread of seasonal flu in sparsely populated areas, where the degree of synchronicity should be lower. For instance, a good case study could be forecasting the seasonal flu in Canada where I expect activity peak times to be much less synchronized.

Also, expanding the analysis to a wider region of the USA, or the whole country would better demonstrate the actual impact of the GMM.

From the literature (see for instance, Charu, Vivek, et al. "Human mobility and the spatial transmission of influenza in the United States." PLoS computational biology 13.2 (2017): e1005382) it is known that human mobility plays a much more relevant role on larger spatial scales than the one analyzed here.

I understand the lack of available surveillance data at country scale and city level is a strong limitation to carry such analysis but some more effort in this direction is necessary to support the use of GMM.

Related to the above comment, I was very curious to see the simulated curves for all NY and NJ counties but I could not find them in the Supplementary Material. I could only find the aggregated HHS Region 2, on Figure 22.

Other two major issues regard the models used to generate the mobility flows used as a benchmark.

The first one is the choice of the gravity model, with a parametrization that is not fully justified. Why the choice of $(d+1)$ to the power of 2, as the denominator? This is a very conservative approach, but in the literature, the exponent is generally fit to the data and can be different than 2.

If the idea was to generate the mobility flows without fitting the parameters, then the radiation model would have been a much better choice.

And this is the second issue: why not using the radiation model as a reference, given that the radiation model has been shown to reproduce commuting flows in the USA extremely well?

Overall, I believe the study brings an important contribution to the field of disease forecasting but it requires a major revision to make the cut in Nature Communications.

Reviewer #2 (Remarks to the Author):

This is a very interesting paper presenting an evaluation of a SEIR model using a mobility model based on data from the Android Google location services. The authors used a very large dataset to build their model. The authors say that the dataset contains 300M users worldwide.

In general, the paper is rigorous in its execution. I think it is difficult any novel modeling contribution. However, it seems to me that the key contribution of this work is indeed the use of the dataset provided by Google to build the mobility model. The key result consists in the fact that the authors are able to obtain prediction results that are similar to that obtained using other "classic" data sources, such as commuters' data. The claim is that data from mobile phones might provide a more up-to-date view of the current mobility patterns of a population. The model is not really global since it is evaluated using data from NYC and neighboring areas.

I have various comments about the manuscript as listed below:

1. My first comment is "philosophical" in nature: the authors say that this model is useful since it provides "real-time" information about users' mobility. However, it seems to me that the model relies on data collected previously. If the mobility patterns change, the model might not be able to capture the current mobility patterns of the population. I believe that this issue should be discussed by the authors.
2. The authors introduce noise in the generation of the map for privacy reasons. I think it would be very useful to measure the impact of the noise, with varying noise level. In fact, researchers should have to set a reasonable level of noise in order that guarantee users' privacy and at the same time lead to useful prediction results. Is there any way of linking the amount of noise introduced and the actual accuracy of the prediction model? This also provides a measure of the robustness of the model in a sense.
3. The authors aggregate the data with a 1-week granularity. It would be interesting to see what happen with different levels of aggregation. Also, why did the authors select 1 week for their analysis? It would also be useful to have an analysis of the stationarity of the flows. In fact, this might have a serious impact on the prediction results for example.
4. A discussion of the bias introduced by considering only Android users would also be extremely useful. For example, the authors should make clear that it is difficult to measure the bias introduced by considering this specific population of users. I think that the authors can do very little in terms of correcting the bias, since it is simply unknown and not measurable. At the same time, some parts of the population are not represented such as infants and children for example.
5. A major source of noise is the fact that the system considered by the authors is not "closed". In fact, for example, there might be people coming from neighboring states or flying into the city. The effect might be limited, but a discussion of these potential issues is important in my opinion.
6. I think the authors should avoid to use "global" in the title and in the text. This is not really a global model. The use of the term "global" is quite misleading in my opinion.

Minor points:

- What are the characteristics of the dataset the authors consider (for example number of users, representativeness of various neighbors of the city?). Are there areas that are underrepresented or overrepresented?
- How did the authors calculate the betweenness? How did the authors extract the network? Why is betweenness interesting in this particular case given the type of mobility model the authors considered.

- How did the authors calculate the correlation between networks? Why is this important? The authors should add a discussion in the paper.
- In Figure 7 the authors say that "the plots are laid out in rough accordance": what do the authors mean with "rough accordance"? More details are needed for reproducibility in my opinion.
- More details about the PatchSim environment would be rather useful (for example, in terms of the actual variables that can be modeled and the parameters that can be set).
- The concept of "patches" is not clearly defined. A more extended definition is needed.
- A clear definition of $\pi(\omega)$ is also needed. Also the authors say that it can be written by "the Bayes theorem" but no details are provided.

Response to Reviewers' Comments

We would like to thank the editor and referees for their valuable comments and suggestions to our manuscript titled 'Forecasting influenza activity using machine-learned mobility map' (NCOMMS-19-03988). In this document, we address their concerns and describe how we have revised the manuscript taking into consideration their inputs. In line with the first reviewer's suggestion, we have extended the scope of experiments by evaluating the dataset and approach on seasonal influenza data from Australia, producing similar results. We have included additional analyses and references in response to concerns raised by the second reviewer. We value reproducibility, open data and code. Therefore, we have obtained approvals to include the underlying mobility map data covering the entire New York City region over the two years studied in the manuscript. As a result, all experiments and results described in the original manuscript can be reproduced and extended with this data and code we will publish on the web.

Reviewer 1:

1. **Comment:** *"[T]he actual benefit of using such data is not so clear ... the claim that 'aggregated mobility data significantly improves the quality of models' does not seem to be supported by the results, while instead appears that the GMM and the commuting data have a nearly identical performance across all forecasting horizons."*

Response: It is indeed true that the GMM based model has very similar performance to that of the model with commuting data. And the fact that it matches the performance is quite useful to know, since GMM's global coverage and spatial resolution allows us to deploy these models in places where commuting data is not available or not regularly updated. We also note that the final results depend on a complex set of interactions that include mobility models, disease propagation models and surveillance data. Further we anticipate additional data will be available with high resolution in the future, paving the way to carry out a more detailed study. As mentioned above we are making our methods and data open to the research community -- we hope this will lead to new findings regarding such datasets and their implications on modeling epidemics. See recent effort by Facebook to make similar (albeit coarser) kinds of data sets available to the researchers <https://data.humdata.org/organization/facebook>

2. **Comment:** *"High resolution mobility could be much more valuable to forecast the spread of seasonal flu in sparsely populated area, where the degree of synchronicity should be lower. A good case study could be forecasting the seasonal flu in Canada where I expect activity peak times to be much less synchronized." "Was curious to see the simulated curves for all NY and NJ counties, but could not find them in the Supplementary material."*

Response: This is an excellent suggestion. We thank the reviewer for hypothesizing that the mobility data can be more valuable in sparsely populated regions. We chose NYC because of the availability of high quality surveillance data, and a reference dataset for commuter flow. We have now added new results for Australia. We chose Australia for the quality of surveillance data and sparsity of population as suggested by the reviewer. We see consistent results in Australia (GMM and COMMUTE perform comparably), and the NO-MOBILITY baseline performs equally well. Additionally, the results demonstrate how national scale modeling can be done using such data sets. With more detailed surveillance data, the approach can be tested in other regions of interest. For reference, we have also added the NY-NJ counties curves to the Supplementary material.

3. **Comment:** *“Expanding the analysis to a wider region of the USA, or the whole country would better demonstrate the actual impact of the GMM. ... it is known that human mobility plays a much more relevant role on larger spatial scales than the one analyzed here.”*

Response: We thank the reviewer for highlighting this reference and raising the possibility of a wider study region of the USA (or the whole country). We considered this possibility ourselves but finally decided to restrict the scope of the present paper so that we can focus on exploring the quality of data and machine learning methods. Additionally, at the larger geographic scales the ground truth surveillance data is more aggregated, which may obscure the impacts of having fine grained mobility data. We presented the current work to be a proof of concept, and as described in response to Comment 2, we chose NYC for the main study, and demonstrated the approach at a larger spatial scale for Australia. The focus of our current paper is on how such a machine-learned mobility map can be used for influenza forecasting. In subsequent papers, we will study the utility of this mobility map across geographic scales ranging from city-level to global models.

4. **Comment:** *“... the choice of the gravity model, with a parametrization that is not fully justified. Why the choice of $(d+1)$ to the power of 2 as the denominator? This is a very conservative approach, but in the literature, the exponent is generally fit to the data”*

Response: We chose the gravity model with inverse square dependence on distance, based on its original unparameterized formulation [1]. $(d+1)$ was added to ensure that the denominator does not become zero for self-loop flows and the uniform addition of unit distance does not affect the results. We did not calibrate the gravity model with commuter data (to estimate the exponent, for example), to ensure that the compared networks are independent of each other. In the Supplementary material (Table 6), we have now included the calibrated coefficients for the unconstrained gravity model with the COMMUTE and GMM datasets.

5. **Comment:** *“If the idea was to generate mobility flows without fitting the parameters, then the radiation model would have been a much better choice. Why not use the radiation model as a reference, given that the radiation model has been shown to reproduce commuting flows in the USA extremely well?”*

Response: We appreciate the reviewer’s suggestion of including a radiation model as a reference. As we understand from the original formulation [2], the radiation model requires the total number of commuters from each patch (i.e., county), which is to be obtained from the commuter dataset (thus violating the independence between the compared models). As mentioned in response to Comment 4, we chose the unparameterized gravity model to ensure that the compared networks are independent of each other. Further, as pointed out in [9], the parameter-free radiation model ignores the effects of spatial scale and heterogeneity, and is known to perform poorly in intra-urban settings (as in our case with NYC) [10].

Reviewer 2:

1. **Comment:** *“the authors say the model is useful since it provides ‘real-time’ information about users’ mobility. However, it seems to me that the model relies on data collected previously. If the mobility patterns change, the model might not be able to capture the current mobility patterns of the population.”*

Response: We would like to highlight that for seasonal influenza in the United States, publicly available surveillance data is delayed by a week or more, for other health conditions the delay is even longer (e.g., Lyme disease on the order of years). By contrast, the GMM is available in near real-time and hence permits running the model to produce *nowcasts* and forecasts. Further, as the disease model is setup, it can use both static and dynamic (time-varying) flow matrix. In order to demonstrate performance and validation against available ground truth, we ran the model retrospectively and hence used past mobility data. We intend to build on this model for real-time forecasting of seasonal influenza in the future. Further, the real-time nature of mobility will be especially useful in nowcasting emerging outbreaks or pandemics.

2. **Comment:** *“I think it would be very useful to measure the impact of the noise, with varying noise level. ... Is there any way of linking the amount of noise introduced and the actual accuracy of the prediction model?”*

Response: We thank the reviewer for the suggestion. The current level of noise added to the mobility data is substantial, in order to preserve the privacy of all individuals involved with a very high epsilon-delta differential privacy bar ($\epsilon=0.66$ and $\delta=2.1e-29$). and yet it produces reasonable performance as presented in the manuscript. It is indeed useful to understand the link between the introduced noise and prediction accuracy (to capture the privacy-performance

tradeoff), but this would require obtaining the anonymized mobility data with varying levels of noise, potentially compromising the differential privacy guarantees. There are ongoing efforts in the information security community [13], to identify the tradeoff between privacy and utility for similar mechanisms for single real-valued query functions, and we believe such efforts will aid in connecting the added noise and forecasting utility of such mobility maps. The effect of perturbations to the underlying flow matrix on the output of a metapopulation model and the resulting performance on forecasting tasks is an interesting research question in its own right.

3. **Comment:** *“The authors aggregate data with 1-week granularity. It would be interesting to see what happens with different levels of aggregation. Why did the authors select 1 week for their analysis? It would also be useful to have an analysis of the stationarity of the flows.”*

Response: Although the ED visit data is available daily, we did notice within-week periodicity which was beyond the scope of our modeling focus and ground truth sample sizes become very small at daily resolution. We chose the 1-week granularity since most real-world influenza surveillance datasets have a weekly granularity. To match this granularity in surveillance, we used the mobility data aggregated to a weekly resolution. Also, observing the mobility data at any finer resolution than a week, may make privacy-preservation challenging. Finally, any coarser granularity (for instance, month) may reduce the number of data points since a typical influenza season lasts about 3-5 months. We have now included autocorrelation plots of the temporal normalized matrix entries in the Supplementary material, which seem to indicate stationarity.

4. **Comment:** *“... discussion of the bias introduced by considering only Android users would also be extremely useful. ... some parts of the population are not represented such as infants and children, for example.” “What are the characteristics of the dataset the authors consider. Are there areas that are underrepresented or overrepresented?”*

Response: It is true that considering only Android users may lead to a sizeable portion of the population not included in the dataset. However, our model makes the assumption that this is fairly representative of the aggregate inter-regional mobility patterns [2,6,7,8]. As recent third party surveys indicate [11,12], Android has a fairly uniform coverage ~60% across gender, ages, and other demographics factors. As we show in Fig. 2a, GMM data correlates fairly strongly with other comparable datasets collected in orthogonal ways. Also we note that any mobility model derived from real world data (for instance commuter flows) will suffer from some form of bias (participation bias in census) -- see [6] for more discussion on the topic.

The novel modeling contribution of this paper is in combining an appropriate noise model for the mobility data (to preserve privacy) with a metapopulation model with suitable parameters (to capture visit rate heterogeneity) to calibrate to emergency department visits. Further, in

calibration we introduce a novel way of parameterizing the seeding of a metapopulation model. We view this work as a **complement**, not a replacement, to existing data sources and techniques. *In principle, every dataset is biased and our long term aim is to statistically model several data sources jointly in a way that reduces this bias.*

5. **Comment:** *“...system considered by the authors is not ‘closed’. ... there might be people coming from neighboring states or flying into the city.”*

Response: We agree with the reviewer’s observation that the system being considered is not closed, due to flows from/to neighboring states or due to international air travel. To look at the impact on neighboring states, we did model the NY-NJ states (HHS Region 2), whose results are presented in Figure 7 (and Supplementary material). Modeling disease dynamics for any region poses this ‘open world’ problem. Finding correction terms, in general, would be a broad open question worth pursuing. To do so in a data-driven fashion, in addition to incoming flows we will also need some indicator of influenza activity in those regions which are beyond the scope of the current work.

6. **Comment:** *“...authors should avoid to use ‘global’ in the title and in the text. This is not really a global model. The use of the term ‘global’ is quite misleading”*

Response: We used the phrase ‘global mobility map’ to highlight the scope of the mobility dataset and its potential impact if proven to be useful in forecasting influenza at a scale where surveillance data is readily available. While in this paper we chose to highlight the high resolution nature of this dataset by doing sub-city level influenza forecasting as a proof of concept, but we also explore spread across states, as in the NY-NJ study. As a further illustration, we have now included Australia as an additional region of study. However, as suggested, we have now removed ‘global’ from the title, and have referred to the dataset in the paper as anonymized or aggregate mobility map (retaining the acronym GMM).

7. **Comment:** *“How did the authors calculate the betweenness? How did the authors extract the network? Why is betweenness interesting... How did the authors calculate the correlation between networks? Why is this important?”*

Response: We used the definition of betweenness for a weighted network (fraction of pairwise weighted shortest paths passing through a node) [3]. The inverse of the normalized flow between a pair of nodes is used as the edge weight. Betweenness centrality is known to be one of the effective heuristics in controlling epidemics on networks [4]. Although the relationship to a metapopulation model is not evident, betweenness is a useful measure to capture critical counties for the mobility flow. The correlation was computed as the Pearson correlation coefficient between the flattened flow matrices (i.e., vectors). It was used to show

the similarity (or lack thereof) between two flow matrices.

8. **Comment:** *“In Figure 7 the authors say that ‘the plots are laid out in rough accordance’. What do the authors mean with rough accordance?”*

Response: By ‘rough accordance’ we meant that the county specific plots are positioned to reflect their geographic adjacency. We have now removed this phrasing to avoid any confusion, and use color coding instead to distinguish the neighboring counties.

9. **Comment:** *“More details are needed for reproducibility.”*

Response: We value reproducibility, open data and code. We have obtained approvals to include the underlying mobility map data covering the entire New York City region over the two years studied in the manuscript. As a result, all experiments and results described in the original manuscript can be reproduced and extended with this data and code we will publish on the web. To our knowledge, this is the first time Google will release dataset of this type and scope. We believe this will go beyond reproducibility and encourage other researchers to study novel questions as regards to human mobility. Our methodology has been described in the main text and Supplementary information section.

10. **Comment:** *“More details about the PatchSim environment would be rather useful... The concept of patches is not clearly defined.”*

Response: PatchSim is a metapopulation SEIR model simulated using difference equations. The exact equations are provided in the Supplementary material Section 2.3. Further, the code, software documentation and model description are available in the git repository [5]. From metapopulation modeling terminology, patches are habitable units (e.g., spatial regions) within which homogeneous mixing of individuals is assumed. For instance, in the NYC study, the individual boroughs (five of them) are the patches, whereas in the Australia study, the 8 states (including Northern Territory and Australian Capital Territory) are modeled as separate patches.

11. **Comment:** *“A clear definition of $\pi(\omega)$ is also needed. Also the authors say that it can be written by the Bayes theorem, but no details are provided.”*

Response: For the parameters of the model, $\pi(\theta)$ represents the prior distribution assumed (for instance, uniform distribution over the range). Given the surveillance data y , one can compute the likelihood of obtaining y given parameter θ . By Bayes theorem, the likelihood and prior are multiplied and normalized to obtain the posterior distribution on the

parameters. More details on our calibration methodology are provided in the Supplementary material Section 2.4.

References:

- [1] Roy J., Thill J.C. Spatial interaction modelling. *Pap. Reg. Sci.* 2003;83:339–361.
- [2] Simini F., González M., Maritan A. & Barabási A. A universal model for mobility and migration patterns. *Nature* 484, 96–100 (2012).
- [3] Newman, M. E. (2001). Scientific collaboration networks. II. Shortest paths, weighted networks, and centrality. *Physical review E*, 64(1), 016132
- [4] Salathé, M., & Jones, J. H. (2010). Dynamics and control of diseases in networks with community structure. *PLoS computational biology*, 6(4), e1000736.
- [5] PatchSim Github repository, <https://github.com/srinivvenkat/PatchSim>
- [6] Barbosa, H., Barthelemy, M., Ghoshal, G., James, C.R., Lenormand, M., Louail, T., Menezes, R., Ramasco, J.J., Simini, F. and Tomasini, M., 2018. Human mobility: Models and applications. *Physics Reports*, 734, pp.1-74.
- [7] Gonzalez, M.C., Hidalgo, C.A. and Barabasi, A.L., 2008. Understanding individual human mobility patterns. *nature*, 453(7196), p.779.
- [8] Song, C., Qu, Z., Blumm, N. and Barabási, A.L., 2010. Limits of predictability in human mobility. *Science*, 327(5968), pp.1018-1021.
- [9] Masucci, A. Paolo, et al. "Gravity versus radiation models: On the importance of scale and heterogeneity in commuting flows." *Physical Review E* 88.2 (2013): 022812.
- [10] Liang, Xiao, et al. "Unraveling the origin of exponential law in intra-urban human mobility." *Scientific reports* 3 (2013): 2983.
- [11] Devices & Demographics, Fluent Annual Survey, 2016
http://www.fluentco.com/wp-content/uploads/2016/01/Fluent2_DevicesandDemographics_2016.pdf
- [12] Devices & Demographics, Fluent Annual Survey, 2017
http://www.fluentco.com/wp-content/uploads/2017/01/Fluent_DevicesandDemographics_2017.pdf
- [13] Quan Geng, Wei Ding, Ruiqi Guo, and Sanjiv Kumar, Privacy and Utility Tradeo in Approximate Dierential Privacy, available online at <https://arxiv.org/abs/1810.00877>, 2019

Reviewers' comments:

Reviewer #1 (Remarks to the Author):

I thank the authors for the extensive revision of their manuscript and their replies to my comments.

Overall, I believe the manuscript has substantially improved and my comments have been adequately addressed.

In particular, I applaud the effort of including the case of Australia which represents a useful example to demonstrate the applicability of GMM to settings where influenza forecasting is more challenging.

Regarding the radiation model, I am still not fully convinced that it would not be a comparable model.

On the one hand, it is true that the RM requires the fraction of outgoing commuters as an input, thus being not fully independent from the commuting data. On the other hand, a simple assumption is that the fraction of outgoing commuters is constant for the whole country thus requiring only the total number of commuters in the country as an input (as done in the original paper by Simini et al.).

Under this assumption, the model is still not fully independent from the commuting data but it becomes a matter of a single global constraint only.

The limitations of the RM in the context of urban areas are true but in this work, the authors are considering US counties, which is precisely the spatial resolution for which the RM has been originally developed.

Therefore, I still believe that an additional forecasting experiment including the radiation model would nicely complete the study.

Reviewer #2 (Remarks to the Author):

The reviewer would like to thank the authors to addressed all the points raised in the previous round of the reviews.

The reviewer has some still some outstanding concerns as listed below:

Comment 1 (use of the model for real-time forecasting): I think that these aspects should be clarified better in the text. It is not completely clear if a GMM model might adapt in case for example of change of behavior due to a pandemic. A typical case might be that people stop traveling or going to places with many people around.

Comment 2 (Impact of noise): I found the explanation provided by the authors satisfactory.

Comment 3 (Type of data aggregation and stationarity of the flow): the comment about the aggregation of the data is satisfactory. Instead, it is not completely clear to me if the temporal normalized matrix might suggest stationarity. In fact, this might be due to the fact that the data are pre-aggregated? What is the impact of this potential non-stationarity? I believe that this should be discussed in the text.

Comment 4 (Representativity of a sample composed of Android users): I wonder if there is a potential impact in terms of under-representation of users with certain socio-economic characteristics. The comments made by the authors are fine, but they should be reported in the paper and listed as limitations.

Comment 5 (Issues related to modeling states as closed systems): the comments provided by the authors are fine, but I wonder if this should be expanded more in the paper: can you model this explicitly by adding a term in the model for describing this? I am not asking to do this explicitly but to discuss this in the paper itself.

Comment 6 (Use of the term "global" in the title): the authors addressed this issue.

Comment 7 (Calculation of betweenness and correlation between networks): I would indicate these references/definitions directly in the text.

Comment 8 (Use of the term "rough accordance"): the modification made by the authors is satisfactory.

Comment 9 (Reproducibility issue): the fact that the authors are going to release the dataset is indeed laudable. The reviewer wonders if the authors are also going to release some of the key parts of the code-base used by the experiments. My worry is that the proposed system might rely on choice of parameters, etc., which are not clearly stated in the text and the related work.

Comment 10 (Details about the PatchSim evaluation): the information provided by the authors is sufficient. The reviewer wonders if it might be a good idea to add these details to the text itself.

Comment 11 (Calculation of $\pi(\omega)$): the clarification provided by the authors is satisfactory.

Response to Reviewers' Comments

We would like to thank the editor and reviewers for their feedback on our revised manuscript titled 'Forecasting influenza activity using machine-learned mobility map' (NCOMMS-19-03988A). We have now addressed the outstanding concerns and have revised the manuscript in accordance with their suggestions.

Reviewer 1:

We thank the reviewer for highlighting the need for comparing against radiation model, and suggesting a way to use a global parameter for deriving the flow matrices. We have now generated the radiation model flows for NYC and Australia and included it in the overall comparison for forecasting performance. We have also presented a statistical comparison of the radiation model based flows with the other networks. We have included the relevant figures in both the main paper and the supplementary material.

We find that RADIATION flows match COMMUTE and AMM (anonymized mobility map) better than the GRAVITY model. The forecast performance across RADIATION, COMMUTE and AMM are very similar, with GRAVITY and NO-MOBILITY performing worse overall. We note that, while RADIATION model is mostly parameter free, the global constant (fraction of outflow) may vary with scale (county vs scale) and across countries. For instance, in our case, while for NYC and NJ we used 0.11 (based on Simini et al.'s original paper), for Australia this constant turned out to be closer to 0.004. Also, since it is based on static population estimates and centroid distances, it may not dynamically adapt to changing trends in mobility and population distribution, thus making AMM a valuable resource for epidemic modeling. We however, note in the discussion that, better estimates of human mobility can be derived by combining multiple models and data sources via an ensemble.

Reviewer 2:

As for Reviewer 2's comments we have now incorporated the salient points from our response to Comments 4, 5, 7 and 10 into the main text. Below, we present our responses to some other concerns raised by the reviewer. Also as further clarification to Comment 9, upon acceptance we will release the model and calibration code, along with input flow matrices and associated parameterizations to reproduce our results in the paper.

1. **Comment:** It is not completely clear if a GMM [*sic*] model might adapt in case for example of change of behavior due to a pandemic. A typical case might be that people stop traveling or going to places with many people around.

Response: For the study reported in the paper, we used the average anonymized mobility map (AMM) since it had similar performance with that of the weekly model, and is a fair comparison with the other static mobility networks. During the case of a pandemic, the weekly AMM is capable of detection in near real-time the alterations to mobility patterns due to disease activity (social distancing, reduced mobility of sick individuals, etc.) better than other static versions of human mobility, and hence may be more relevant.

2. **Comment:** It is not clear if the temporal normalized matrix might suggest stationarity. In fact, this might be due to the fact that the data are pre-aggregated? What is the impact of this potential non-stationarity? This must be discussed in the text.

Response: It is true that the normalized matrix may not suggest stationarity of the raw flows. We evaluated the stationarity of the normalized versions since these serve as the input to the metapopulation model, and hence more relevant for our experiments.

3. **Comment:** I wonder if there is a potential impact in terms of under-representation of users with certain socioeconomic characteristics. The comments made by the authors are fine, but they should be reported in the paper and listed as limitations.

Response: Since the mobility map mainly covers users with smartphones, some socioeconomic and demographic groups will be under-represented. Such sampling bias exists invariably in any dataset, including the disease surveillance. We have included our response in the main text as part of the discussion.

4. **Comment:** (Issues related to modeling states as closed systems): I wonder if this should be expanded more in the paper: can you model this explicitly by adding a term in the model for describing this? I am not asking to do this explicitly but to discuss this in the paper itself.

Response: Indeed, one can address the open-world phenomenon by adding an external parameter. In fact, one could simulate a national or even global model given the coverage of the dataset. We modeled it as a closed system to avoid issues of overfitting (through additional parameters) and lack of standardization in tackling this issue. Our results show that this assumption does not significantly affect the models' performance. We have now addressed this point in the discussion section of the paper.

REVIEWERS' COMMENTS

Reviewer #1 (Remarks to the Author):

I commend the hard effort made by the authors to address my remarks. I think that including the radiation model in their work has made their analysis substantially stronger and exhaustive.

Overall, this is an important manuscript that advances the field of human mobility modeling.

I recommend the paper to be accepted for publication in Nature Communications.

Reviewer #2 (Remarks to the Author):

I went through the changes in the manuscript and found them satisfactory.

I would recommend this paper for publication.